# Citizen science improves our understanding of the impact of soil management on wild pollinator abundance in agroecosystems

**Logan R. Appenfeller** [ORCID]*, **Sarah Lloyd, Zsofia Szendrei**

Department of Entomology, Michigan State University, East Lansing, Michigan, United States of America

* appenfel@msu.edu

## Abstract

Native bees provide essential pollination services in both natural and managed ecosystems. However, declines in native bee species highlight the need for increased understanding of land management methods that can promote healthy, persistent populations and diverse communities. This can be challenging and costly using traditional scientific methods, but citizen science can overcome many limitations. In this study, we examined the distribution and abundance of an agriculturally important wild bee species, the squash bee (*Eucera* (*Peponapis*) *pruinosa*, Hymenoptera: Apidae). They are ground nesting, specialist bees that depend on cultivated varieties of *Cucurbita* (squash, pumpkins, gourds). The intimate relationship between squash bees and their host plants suggests that they are likely sensitive to farm management practices, particularly those that disturb the soil. In this study, citizen scientists across Michigan used a survey to submit field management and bee observation data. Survey results indicated that squash bees occupy a wide geographic range and are more abundant in farms with reduced soil disturbance. Citizen science provided an inexpensive and effective method for examining impacts of farm management practices on squash bees and could be a valuable tool for monitoring and conserving other native pollinators.

## Introduction

Pollinators are important in both natural and managed ecosystems to maintain plant genetic diversity, contribute to ecosystem stability, and sustain crop production [1]. Of the most commonly produced crops globally, 35% rely on or benefit from animal pollination which is provided mostly by insects such as western European honey bees (*Apis mellifera*, Hymenoptera: Apidae) and a wide variety of wild, native bees [1,2]. Honey bees are the most prolific pollinators of pollinator dependent crops [3], however annual losses of managed honey bees can currently reach as high as 50% due to a suite of factors such as exposure to pesticides, reduced forage availability, parasites, and diseases [4,5]. As a result, researchers are investigating the role of wild bees as crop pollinators, which are declining due to human disturbances such as habitat loss/fragmentation [6], landscape simplification [7], and increased pesticide use [8]. Management practices that increase abundance and species richness of native bees can

**Data Availability Statement:** All data and R code files are available from the Dryad database (DOI: https://doi.org/10.5061/dryad.4qrfj6q69), (Reviewer URL: https://datadryad.org/stash/share/

W-GYVqNY7m_Cv6qMu6srShpdTm_
FaDngFHm9_sfy7QM). The data and R codes were
submitted to Dryad under the option "Private for
Peer Review". The Reviewer URL link above
provides reviewers with access to all data and R
code.

**Funding:** Z.S. 2016-51300-25732 United States
Department of Agriculture National Institute of
Food and Agriculture https://nifa.usda.gov The
funders had no role in study design, data collection
and analysis, decision to publish, or preparation of
the manuscript.

**Competing interests:** The authors have declared
that no competing interests exist.

ameliorate crop pollination deficiencies [9], especially for crops that are more effectively pollinated by native bees [10–15].

Studying changes in insect populations is often challenging [16,17], and in order to collect baseline abundance and distribution data, insect monitoring has in some cases turned to citizen science as an effective method for gathering large datasets across broad geographic areas with low costs compared to traditional methods [18,19]. Although citizen science can suffer from limitations such as data accuracy and participant retention, these issues can be negated with proper planning and participant training as demonstrated by many successful citizen science projects. For example, citizen science can be an effective method for monitoring native bees [20–22]. Citizen scientist observations can describe bee species dynamics as well as specimens collected by professional researchers [23], provide data on specific aspects of bee biology, including the nesting habits of solitary bee species [24], and the impacts of flowers and surrounding natural land cover on plant-bee interactions [25]. Participants in pollinator citizen science projects often volunteer because of a desire to learn about bees and to contribute to science [26]. This provides opportunities for large-scale, cost-effective studies that simultaneously allow scientists to educate the public about ecological issues such as the loss of biodiversity. Actively engaging with the public through hands-on experiences provides more impactful education that can enhance learning and inspire continued action [27].

Our study focused on squash bees (*Eucera* (*Peponapis*) *pruinosa*, Hymenoptera: Apidae), an important specialist pollinator of *Cucurbita* (e.g. pumpkins, squash, gourds). This plant genus is dependent on pollination and is an ideal system to promote native bees because of their mutualism with squash bees. These specialist bees forage for nectar and pollen on *Cucurbita* flowers, rest within closed flowers, and excavate nests in the soil around *Cucurbita* plants [28–30]. The intimate relationship between squash bees and their host plants indicates that squash bees are potentially sensitive to farm management practices, especially those that manipulate the soil. However, studies examining the relationship between squash bees and farm management practices have produced differing results. For example, tillage can destroy squash bee nests, reduce the number of surviving offspring, alter sex ratios and emergence timing [31], and reduce squash bee flower visitation [32]. Conversely, another study observed similar adult squash bee abundance in tilled and untilled pumpkin fields, and squash bees preferred to nest in irrigated soil near host plants regardless of whether or not the soil was tilled [33]. However, more recent findings suggest that squash bees prefer to nest in tilled soil [34]. Mulching is another ground management practice commonly used in *Cucurbita* production that may deter or inhibit squash bee nesting. Although, previous attempts to compare squash bee nesting frequency in bare soil and soil covered by different types of mulch were inconclusive due to low sample size [35].

Although multiple studies have examined the impacts of farm management on squash bees, the scope of investigation has often been limited to one management practice at a time, sample sizes have been relatively low, and results have often been mixed [31–34]. Here, we used a citizen science survey to determine how squash bee abundance varies according to multiple farm management practices including tillage type, depth, and mulch, and ascertain the distribution of squash bees in Michigan. Citizen science allowed us to increase sampling while providing opportunities to spread awareness among the public about the importance of squash bees which may pollinate about two-thirds of squash grown commercially in the United States [36]. Previous research indicates that citizen science projects are more successful if the participants have prior interest in the subject matter [26,37,38] thus we recruited Michigan State University (MSU) Extension Master Gardeners because of their interest in agriculture, their level of scientific knowledge, and their commitment to educate others in their communities.

## Materials and methods

### Citizen scientist recruiting and training

Master Gardeners were contacted by the program coordinator via email and recruited to participate in our squash bee survey. To train Master Gardeners we invited them to webinar presentations held in June 2017 and 2018 where the methods, project goals, and preliminary results were discussed. Educational workshops (~3h) were held for participants at several locations throughout Michigan in July 2017 (Novi, MI: July 20; Holland, MI: July 21), 2018 (Mason, MI: July 16; Novi, MI: July 20; Grand Rapids, MI: July 25; Lincoln, MI: July 27), and 2019 (Novi, MI: July 18; Grand Rapids, MI: July 25). Each workshop included a classroom presentation during which participants were taught about the biology of cucurbit flowers, squash bees, bee identification, the importance of native pollinators, and the methods for collecting data and submitting surveys. Master Gardeners were provided with supplemental educational materials including a factsheet with information pertaining to the pollination system of cucurbits and their relationship with squash bees, and a brief bee identification guide. Presentations were followed by an outdoor session in a squash garden or farm where participants practiced identifying squash bees at flowers and filling out the squash bee survey (Fig 1).

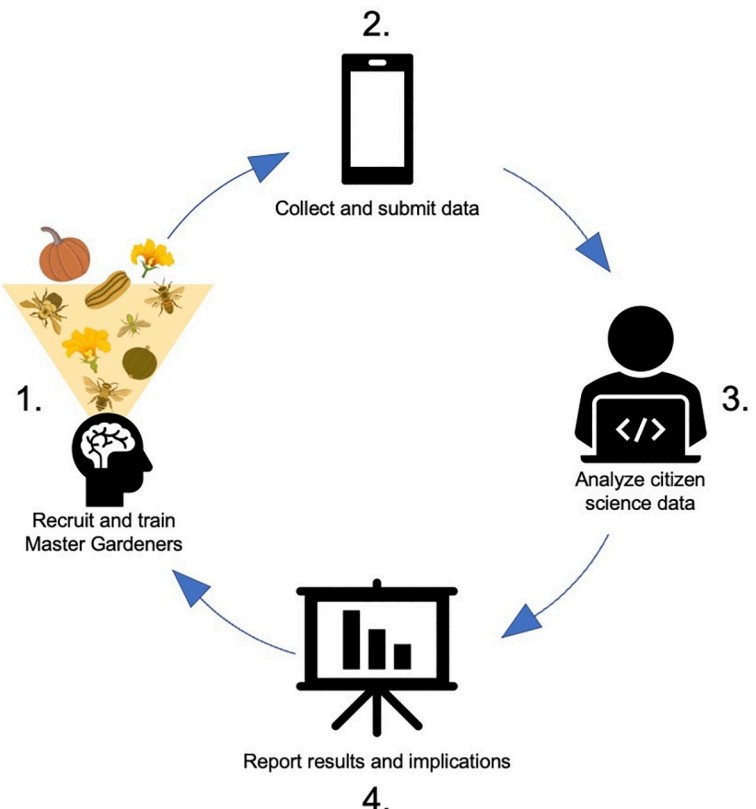

**Fig 1. Conceptual diagram illustrating the process of training, data collection, data analysis, and reporting for this citizen science project.** Michigan State University Extension Master Gardeners were taught about the pollination system of cucurbits, the importance of squash bees, and how to identify bees visiting squash flowers (1). Master Gardeners collected data on squash bees which they submitted using the Squash Bee Survey smartphone application (2). Surveys could also be submitted through a web browser or via paper copies. Data was analyzed and verified with photos submitted by participants (3). Results were shared with Master Gardeners via webinars, presentations, and a factsheet (4).

## Squash bee survey and observation protocol

The primary method used for data collection and survey submission was the Squash Bee Survey smartphone application (Fig 2) developed in the MSU Vegetable Entomology Lab using Google forms [39] and AppSheet [40]. Surveys were also made available for participants in printable PDF and web browser versions accessible through the MSU Vegetable Entomology website (https://vegetable.ent.msu.edu). In each survey, Master Gardeners provided the last four digits of their phone number as unique, confidential identifiers, and were asked several questions pertaining to the location and management of the farm where they conducted surveys (S1 Fig). Tillage type was one of the primary factors of interest in our study and participants could select no tillage, reduced tillage, or full tillage. No tillage is characterized by a lack of soil disturbance between harvesting and planting crops resulting in the presence of crop stubble or residues. Reduced tillage (a.k.a. conservation tillage) is defined by lower tillage intensity resulting in the retention of some crop residues on the soil surface. Both of these methods help to prevent soil erosion, increase water retention, and conserve energy resources. Full tillage (a.k.a. conventional tillage) uses cultivation (e.g. ploughing, harrowing) as the primary means of weed control and seedbed preparation resulting in a loose soil surface and lack of plant residues on the soil surface [41]. Tillage depth (0 cm, 3–14 cm, 15–25 cm), and mulching practices (none, plant material, plastic) were also of interest, and participants selected all categories that represented the practices used in a particular crop field. Surveys submitted electronically via the smartphone application or web browser option were automatically entered into a Google Sheets spreadsheet with a timestamp and stored in Google Drive via AppSheet [40]. Printed surveys received by mail were entered into the spreadsheet manually upon receipt. There was no limit to the number of surveys each participant could submit.

Master Gardeners were asked to conduct bee surveys on cucurbit flowers in the morning (~07:00–12:00) while squash bees were active, on sunny days with no more than light winds

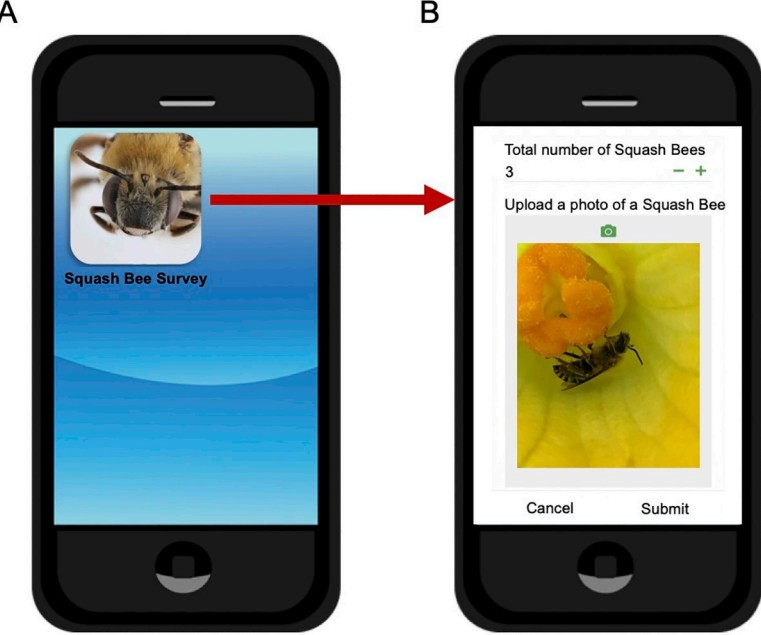

**Fig 2.** Squash bee survey smartphone application installed on a smartphone (A). Screenshot from the squash bee survey (B). The smartphone application was the primary platform for Master Gardeners to submit information about the bees observed in cucurbit flowers and management practices used on farms.

and air temperatures of at least 21 ºC. For each survey, five separate cucurbit flowers were observed for 1 min each for a total of 5 min of observations, and the numbers of squash bees, honey bees, bumble bees, and other bees (any other type of bee) observed visiting the flowers were summed and recorded. At the end of the survey, participants were asked to take a picture of a bee that they identified as a squash bee to be submitted with their electronic data. These photos were used to assess participants' squash bee identification accuracy; photo verifications were not performed for surveys that were submitted by mail. IRB Number is x17-688e; i054192; this survey was deemed exempt and was not subjected to review by an institutional review board or ethics committee.

## Statistical analysis

Squash bee observations were mapped by county and compared to previous county records of this species [42]. The number of surveys and the number of different people participating were calculated for each year. The proportions of squash bees, honey bees, bumble bees, and other bees were calculated to identify the most common type of bee observed during surveys. Generalized Linear Mixed Models using Laplace approximation and negative binomial distribution were used with the 'glmmadmb' function in the 'glmmADMB' package [43] to determine the effects of various management practices on the number of squash bees observed during surveys. Each model contained a single fixed effect (tillage type, tillage depth, mulch, irrigation, insecticides, farm area devoted to cucurbits, type of vine crop observed; S1 Fig) with date nested within county as random effects. Models were individually compared to a null model using the 'anova' function in the 'stats' package [44]. The 'AICctab' function in the 'bbmle' package [45] was used to compare models based on AIC (Table 1). The 'emmeans' function in the 'emmeans' package [46] with the 'fdr' p-adjustment method was used to determine pairwise differences between factor categories for models that differed significantly from the null model.

Kruskal-Wallis tests were performed using the 'kruskal.test' function in the 'stats' package [44] to determine the effects of the previously mentioned management practices on honey bees, bumble bees, and other bees ($\alpha = 0.05$). This analytical method was used for these bee categories due to non-convergence of the Generalized Linear Mixed Model method used for squash bee analyses. Surveys submitted with incorrectly identified squash bee photos, factor categories with less than 5 responses, and numeric outliers (bee counts greater than the third quartile plus 1.5 times the interquartile range for each respective bee category) were excluded from analyses. Squash bees are known to forage solely on species in the genus *Cucurbita*

**Table 1. Akaike Information Criterion (AIC) comparisons of Generalized Linear Mixed Models testing the effects of different cucurbit management practices in a citizen science survey, 2017–2019.** Fixed effects were compared using the difference in AIC (ΔAICc) between the model of the lowest AIC and all other models. AIC weight indicates the probability that a model best describes the data. The model with the lowest ΔAICc and the highest AIC weight is assumed to better fit the data than other models.

| Fixed Effect | ΔAICc | df | Weight |
|---|---|---|---|
| Tillage Type | 0.0 | 6 | 0.858 |
| Cucurbit Area | 7.0 | 5 | 0.026 |
| Null | 7.0 | 4 | 0.026 |
| Irrigation | 7.2 | 6 | 0.023 |
| Mulch | 7.3 | 7 | 0.022 |
| Vine crop observed | 7.6 | 7 | 0.019 |
| Tillage depth | 8.3 | 6 | 0.014 |
| Insecticides | 8.7 | 6 | 0.011 |

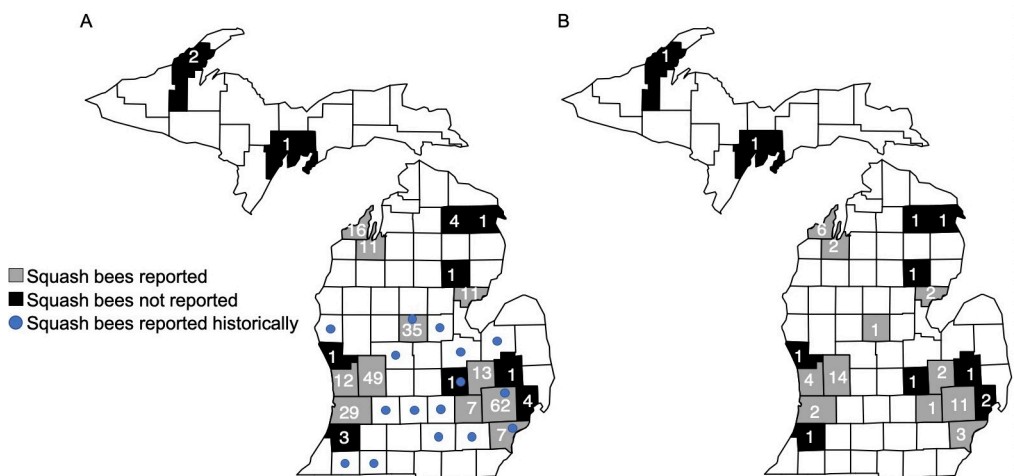

**Fig 3. Squash bees reported in Michigan counties.** Counties from which squash bee surveys were received, the number of surveys submitted from each county, and previous county records of squash bees (A). The number of different participants per county (B). A total of 5 surveys were received from 2 different participants in Floyd Co. Indiana, both of which reported squash bees (Floyd Co. Indiana not displayed on map).

[29,47] consequently, surveys observing only cucumber or melon flowers (*Cucumis*) were also excluded from analyses. When Kruskal-Wallis tests were significant, post hoc analyses were conducted using the 'dunn.test' function in the 'dunn.test' [48] package with the 'holm' p-adjustment method to control family-wise error rates, to determine differences in bee abundances among groups for factors with more than 2 groups ($\alpha < 0.05$). All analyses were performed using R version 3.5.1 [44].

# Results

Of the 291 surveys received, 276 (2017: 56 electronic surveys; 2018: 70 electronic and 7 print surveys; 2019: 101 electronic and 42 print surveys) were used for analysis from 21 Michigan counties and 1 Indiana county. Eleven out of 21 Michigan counties reported observing squash bees (Fig 3A). Of the 11 Michigan counties that reported squash bees, only four overlapped with historical reports [42], and the remaining seven provide new county records. A total of 59 people participated in this study (Fig 3B), 87% of whom submitted observations from organic farms, with some participating in multiple years (2017: 19 different participants; 2018: 27 different participants; 2019: 23 different participants). Out of all surveys, 48% included photos, 90% of which were correctly identified as squash bees. Squash bees accounted for 51% of bees reported over the combined 3 years (Fig 4) and were the most common type of bee reported in each year (2017: 67%, 2018: 33%, 2019: 54%).

## Effects of management practices on squash bees

The number of squash bee visits reported per survey varied according to tillage type ($\chi^2 = 11.18$, df = 2, $p < 0.01$; Fig 5). The mean number of squash bees reported in farms using no tillage (mean = $2.86 \pm 0.27$ (SE)) was more than 3 times greater than the mean number in full tillage (mean = $0.92 \pm 0.34$ (SE); p = 0.02), but only slightly greater than in reduced tillage (mean = $2.55 \pm 0.26$ (SE); p = 0.78). The mean number of squash bees reported in reduced tillage farms was about 3 times greater than in those using full tillage (p = 0.02).

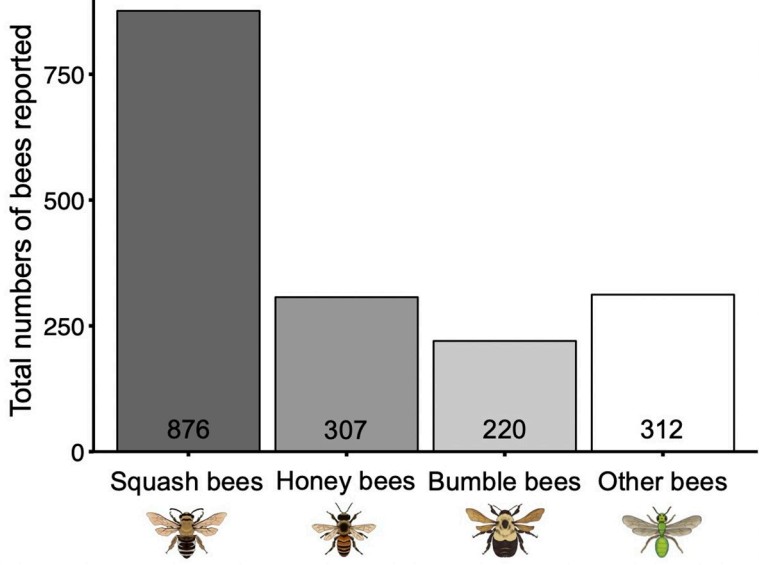

**Fig 4. Types of bees reported by citizen scientists.** Total numbers of squash bees, honey bees, bumble bees, and other bees observed by citizen scientists visiting cucurbit flowers over the summers of 2017, 2018, and 2019 combined.

Tillage depth did not affect squash bee visitation ($\chi^2 = 2.91$, df = 2, p = 0.23; Fig 6). Likewise, squash bee visitation did not significantly vary by mulch type ($\chi^2 = 5.98$, df = 3, p = 0.11) (Fig 7). However, the mean number of squash bees reported in the 'Plastic' (mean = 4.50 ± 1.67 (SE)) and 'Plastic + Plant Material' (mean = 3.86 ± 1.18 (SE)) groups were more than 1.5 times greater than in both the 'None' (mean = 2.46 ± 0.30 (SE)) and 'Plant Material' (mean = 2.36 ± 0.20 (SE)) groups.

Squash bee visitation was not affected by insecticides ($\chi^2 = 2.53$, df = 2, p = 0.28), irrigation ($\chi^2 = 3.97$, df = 2, p = 0.14), the type of vine crop observed ($\chi^2 = 5.68$, df = 3, p = 0.13), or

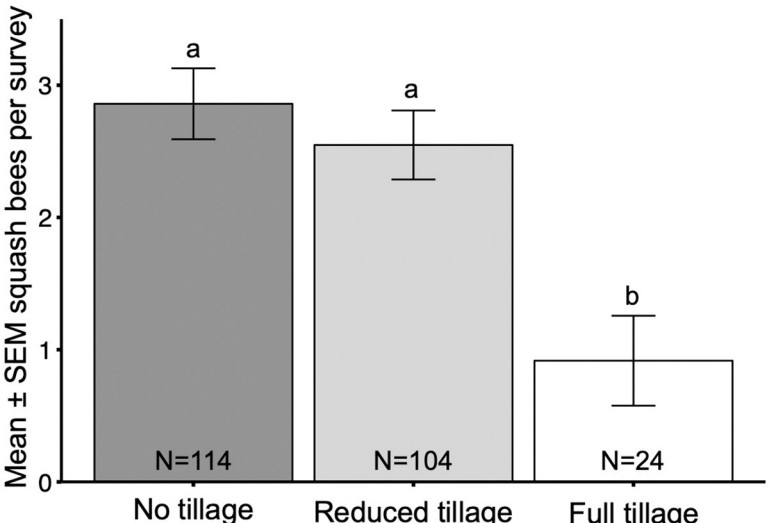

**Fig 5. Squash bees by tillage type.** Mean ± SEM number of squash bees reported in a squash bee survey conducted by citizen scientists in farms using different types of tillage, combined for 3 years (2017, 2018, 2019).

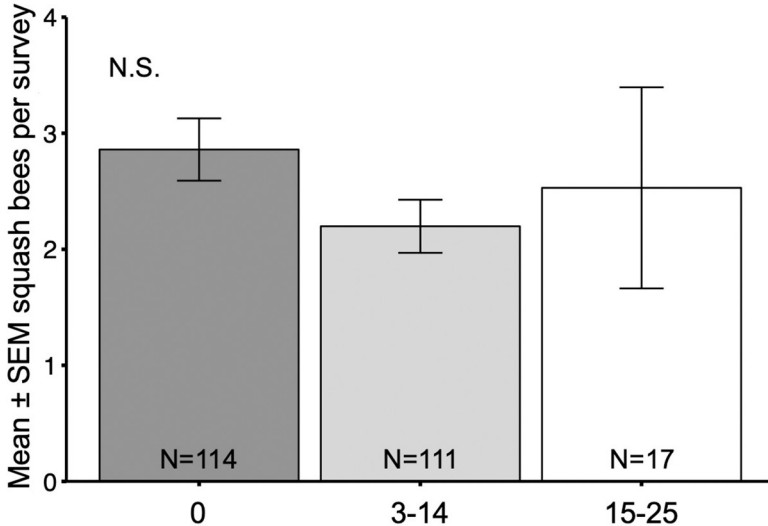

**Fig 6. Squash bees by tillage depth.** Mean ± SEM number of squash bees reported in a squash bee survey conducted by citizen scientists in farms using different tillage depth (cm), combined for 3 years (2017, 2018, 2019).

amount of area devoted to cucurbit growth ($\chi^2$ = 2.10, df = 1, p = 0.15). Based on AIC model comparison, tillage type ($\Delta$AICc = 0.0, df = 6, weight = 0.858) explains the patterns in squash bee visitation better than other analyzed factors (Table 1).

## Effects of management practices on honey bees, bumble bees, and other bees

No relationship was observed between honey bee visitation and tillage type ($\chi^2$ = 1.40, df = 2, p = 0.50), tillage depth ($\chi^2$ = 1.05, df = 2, p = 0.59), insecticides ($\chi^2$ = 1.13, df = 2, p = 0.57), irrigation ($\chi^2$ = 0.30, df = 2, p = 0. 86), or the type of vine crop observed ($\chi^2$ = 6.50, df = 3,

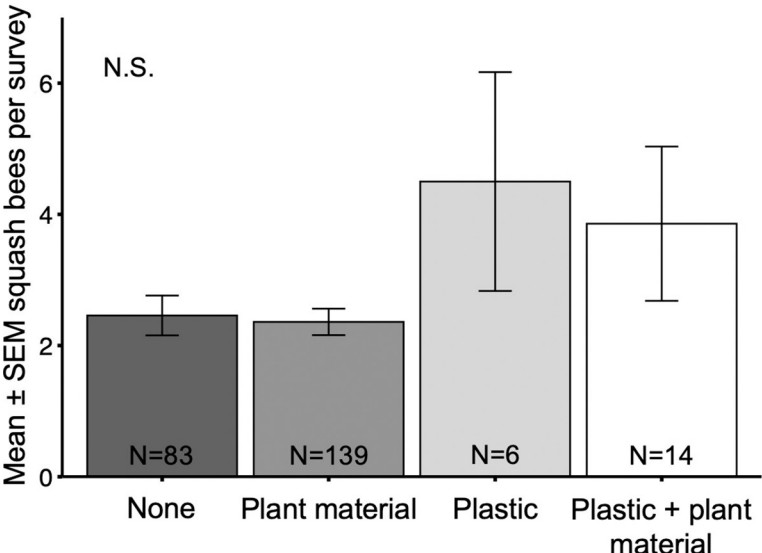

**Fig 7. Squash bees by mulch type.** Mean ± SEM number of squash bees reported in a squash bee survey conducted by citizen scientists in farms using different mulching practices, combined for 3 years (2017, 2018, 2019).

p = 0.09; S1 Table). However, mulch affected honey bee visitation ($\chi^2$ = 12.02, df = 3, p = 0.01). The mean number of honey bees reported in farms using plant material mulch (mean = 0.44 ± 0.06 (SE)) was more than 3 times greater than when mulch was not present (mean = 0.13 ± 0.04 (SE); p < 0.01). No other significant differences were observed among mulch types, however, the mean number of honey bees observed when plant material and plastic (mean = 0.33 ± 0.14 (SE)) or plastic alone (mean = 0.40 ± 0.40 (SE)) were present was approximately 2.5–3 times greater than when mulch was not present (S1 Table). Average honey bee visitation was also 2 times greater when cucurbit area was greater than 0.4 hectares (mean = 0.64 ± 0.24 (SE)) compared to less than 0.4 hectares (mean = 0.31 ± 0.04 (SE); $\chi^2$ = 2.92, df = 1, p = 0.09; S1 Table). No relationship was observed between bumble bee visitation and tillage type ($\chi^2$ = 2.96, df = 2, p = 0.23), tillage depth ($\chi^2$ = 5.05, df = 2, p = 0.08), mulch ($\chi^2$ = 1.56, df = 3, p = 0.67), insecticides ($\chi^2$ = 3.62, df = 2, p = 0.16), or irrigation ($\chi^2$ = 4.13, df = 2, p = 0.13). Interestingly, bumble bee visitation was 3.5 times greater when cucurbit area was less than 0.4 hectares (mean = 0.54 ± 0.05 (SE)) compared to greater than 0.4 hectares (mean = 0.15 ± 0.10 (SE); $\chi^2$ = 3.60, df = 1, p = 0.06). Bumble bee visitation also varied by the type of vine crop observed ($\chi^2$ = 20.56, df = 3, p < 0.01). The mean number of bumble bees observed visiting winter squash varieties (*C. pepo*, mean = 0.80 ± 0.09 (SE)) was more than two times greater than when mixed varieties (combination of *C. pepo* and *Cucumis*; mean = 0.35 ± 0.12 (SE); p < 0.01) or summer squash varieties alone (*C. pepo*, mean = 0.36 ± 0.06 (SE); p < 0.01) were observed. No other significant differences were found among vine crop types; however, the mean number of bumble bee visits in surveys where summer and winter squash flowers were observed together (mean = 0.70 ± 0.18 (SE)) was approximately 2 times greater than when mixed or summer squash flowers alone were observed. Other bee visitation did not vary by mulch ($\chi^2$ = 4.26, df = 3, p = 0.23), insecticides ($\chi^2$ = 4.77, df = 2, p = 0.09), irrigation ($\chi^2$ = 0.58, df = 2, p = 0.75), or the type of vine crop observed ($\chi^2$ = 6.07, df = 3, p = 0.11; S1 Table). Although, visitation by other bees varied according to tillage type ($\chi^2$ = 11.04, df = 2, p < 0.01) with the mean number of other bees observed in farms using no tillage (mean = 1.33 ± 0.14 (SE)) and full tillage (mean = 1.36 ± 0.32 (SE)) being approximately 2 times greater than in farms using reduced tillage (mean = 0.68 ± 0.09 (SE); p < 0.01; p = 0.16). Tillage depth also affected other bee visitation ($\chi^2$ = 8.66, df = 2, p = 0.01) with the mean number of other bee visits in farms using no tillage (mean = 1.33 ± 0.14 (SE)) being around 1.5 times greater than in farms with 3–15 cm tillage depth (mean = 0.78 ± 0.10 (SE); p = 0.01) or 15–25 cm (mean = 0.96 ± 0.24 (SE); p = 0.71; S1 Table).

## Discussion

This study was the first to successfully use citizen science to gather a large dataset to examine an agriculturally significant native bee's distribution and to determine how their flower visitation frequency varies according to crop management. Some of the successes of our project are due to identifying an appropriate target audience for involvement with the project, a thorough volunteer education process, simplicity of the survey protocol, and ease of data submission. These allowed us to not only sustain the project for 3 years, a longer duration than many other pollinator citizen science projects [23,49,50], but increase data collection in each project year. Pollinator citizen science projects with more complex, time consuming experiments tend to have problems retaining or increasing participant numbers [51], which was one of the reasons for keeping our protocols relatively simple. For example, we simplified the data collection methods by asking participants to count bees at flowers instead of using bee nest count data which would have been a more direct measure of the impact of soil management practices, but this would have required more time and effort from citizen scientists. Furthermore, nesting data would be prone to error and difficult for us to verify via photos. Since squash bees spend

the majority of their time in *Cucurbita* flowers and tend to nest close to their host plants [33], flower visitation frequency is likely directly related to overall squash bee abundance. In addition to adjusting sampling methods to the ability level of participants, incorporating technological advances were also important for success as has been demonstrated in previous citizen science studies [52]. We used smartphones as our primary method for data collection and photo submission as many of our citizen scientists were familiar with this technology. To keep the survey relatively short, we omitted some questions that would have provided us with valuable data, for example, recording soil-type, time of day, weather conditions, flower sex, and flower abundance would have allowed us to answer additional questions. Despite these limitations, we find that citizen scientists are eager to be involved with these types of data collection efforts and that they can contribute valuable information to science.

Our survey focused mainly on soil management methods because of the need to better understand their intimate interactions with ground-nesting bees. As soil conservation methods, such as strip-tillage, gain more acceptance in agriculture [53–55], their impacts on beneficial arthropods need to be evaluated. The amount of area and depth of soil disturbance as well as mulching practices were our primary interests, since these are likely to destroy squash bee nests or interfere with nesting behavior. Survey results suggest that on average, flowers in non-tilled farms received approximately three times more squash bee visits than when full tillage was used. This is concurrent with previous study results that also found increased squash bee flower visitation [32] and offspring emergence [31] when soil was not tilled. Additionally, surveys of flowers on reduced tillage farms reported only slightly fewer average squash bee visits than no till surveys which indicates that both of these practices can contribute to squash bee population conservation at similar levels.

It is possible that tillage is correlated with other types of management practices that are responsible for changes in squash bee abundance, such as crop rotation or insecticide use. Considering that squash bees nest close to their host plants [33] and that *Cucurbita* crops are typically rotated, the number of squash bees visiting flowers is likely influenced by the management of previous year's fields, and the distance between these and current plantings. This highlights the relevance of ground management not only within individual fields but at the farm level. However, previous research demonstrating significant impacts on squash bee abundance due to soil management combined with a lack of observed impacts on generalist pollinators like honey bees and bumble bees suggests that although other forms of farm management may have some impact on squash bees, soil tillage is likely to impose strong effects [32].

Strip-tillage is often accompanied by the presence of cover crop residues (mulch) between strips of tilled soil which can help maintain soil moisture, reduce soil erosion, and inhibit weed seed germination [56]. We did not observe a significant decrease in average squash bee visitation where mulch was present, but rather a numerical increase. Although this increase was not statistically significant, overall, mulch did not appear to inhibit or deter squash bees from visiting flowers. This is an important finding since we expected that mulch may deter females from digging nests, which are typically observed in bare soil [28]. Conversely, our results suggest that squash bees may successfully build nests despite the presence of mulch which is similar to previous observations where squash bees nested in vegetated soil [57].

Surprisingly, tillage depth had no observed effect on squash bee flower visitation although we expected that shallower tillage may lead to increased squash bee visitation due to conservation of nests. It is possible that although deeper tillage destroys more squash bee nests it is not directly related to bee numbers counted at flowers if bees nest in the field perimeters where they are protected from soil disturbance. In addition, it may have been difficult for some citizen scientists to accurately approximate tillage depth, resulting in mis-categorizations. However, considering squash bee abundance was significantly lower in fully tilled fields, the

amount of tilled area may have a greater impact than the depth of tillage. Our results suggest that in crops where tillage is necessary, reduced tillage can provide similar levels of native soil nesting bee conservation compared to no tillage.

Citizen science was also an effective means of examining the current geographic range of the squash bee in Michigan because of the relatively broad participation in our study. Citizen scientists reported 7 new county records for this species, and while geographic range expansion may be responsible for such patterns, we hypothesize that a more likely explanation is a lack of historical reports and/or an increase in the number of small organic farms in Michigan which more often practice farm management methods that can promote native bees [58–60].

In our survey, squash bees were observed visiting flowers about 3–4 times more often than honey bees, bumble bees, or other bees, comprising more than 50% of the total number of bees reported (Fig 4). We did not expect that honey bees would respond to mulching or tillage because they are not ground nesting bees and therefore do not directly interact with the soil. We observed a positive effect of plant mulches on honey bee abundance which may be due to an indirect impact of soil management practices on these bees through affecting plant health or flower abundance. Bumble bees did not respond to ground management practices, and although they are ground nesting bees, they can cover long distances during foraging and are likely nesting outside of squash fields [32,61–63]. The overall lack of response by honey and bumble bees to most soil management practices could also be because they are dietary generalists feeding on other available sources of pollen and nectar [64]. Additionally, honey bees in particular visit squash flowers primarily for nectar as indicated by their preference for pistillate squash flowers [65]. Therefore, squash flower abundance, quality, and/or field attributes dictated by soil management may be less consequential for these bees.

In summary, implementing management practices such as reduced tillage can help conserve native bees by providing suitable nesting habitat and allow farmers to take advantage of natural pollination services. Declines in both native and managed bees highlight the need to increase these types of conservation efforts [66–68] and non-traditional scientific methods like citizen science can provide new solutions. Despite its limitations, citizen science has proven to be an effective tool and it should be utilized when possible due to its ability to yield large amounts of quality data and provide citizens with an impetus for action towards issues like native pollinator conservation.

## Supporting information

**S1 Fig. Squash bee survey filled out by citizen scientists, 2017–2019.** Surveys were submitted via a smartphone application, web browser, or mail.
(PDF)

**S1 Table. Data summary and results of statistical analyses for honey bees, bumble bees, and other bees.** Number of observations by cucurbit management method, mean ± SEM, and Kruskal-Wallis test results for honey bees, bumble bees, and other bees in a citizen science survey conducted in 2017–2019. Significant pairwise differences among factor levels were determined using Dunn's test and are indicated by different letters following mean ± SEM values (α < 0.05).
(DOCX)

## Acknowledgments

We are grateful to many individuals for their contributions to this research. MSU Extension Master Gardeners submitted surveys, and Mary Wilson, the MSU Master Gardener Program

Coordinator, recruited Master Gardeners and helped set up educational events. Several of our peers provided valuable feedback on previous versions of the manuscript. Bees in Fig 4 were illustrated by Rebecca Schwutke (squash bee, honey bee, bumble bee) and Laramie Appenfeller (other bee).

## Author Contributions

**Conceptualization:** Sarah Lloyd, Zsofia Szendrei.

**Data curation:** Logan R. Appenfeller, Sarah Lloyd, Zsofia Szendrei.

**Formal analysis:** Logan R. Appenfeller, Zsofia Szendrei.

**Funding acquisition:** Zsofia Szendrei.

**Investigation:** Logan R. Appenfeller, Zsofia Szendrei.

**Methodology:** Logan R. Appenfeller, Sarah Lloyd, Zsofia Szendrei.

**Project administration:** Logan R. Appenfeller, Sarah Lloyd, Zsofia Szendrei.

**Resources:** Zsofia Szendrei.

**Supervision:** Zsofia Szendrei.

**Validation:** Logan R. Appenfeller, Zsofia Szendrei.

**Visualization:** Logan R. Appenfeller, Zsofia Szendrei.

**Writing – original draft:** Logan R. Appenfeller, Zsofia Szendrei.

**Writing – review & editing:** Logan R. Appenfeller, Zsofia Szendrei.

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
