## [Decision Letter · Decision Letter 0]

5 Feb 2020

PONE-D-19-34559

Citizen science improves our understanding of native pollinator management

PLOS ONE

Dear Mr. Appenfeller,

Thank you for submitting your manuscript to PLOS ONE. After careful consideration, we feel that it has merit but does not fully meet PLOS ONE’s publication criteria as it currently stands. Therefore, we invite you to submit a revised version of the manuscript that addresses the points raised during the review process.

From the academic editor:

Thank you for your patience with the review process. As you can see below, both reviewers thought this was a well-written manuscript describing a valuable and interesting study. I think most of the reviewers' comments will improve the usefulness of the paper, mainly through some changes or additions to emphasis. While I think all of the comments are valuable, I specifically agree with Rev. 2 that you should be cautious in evoking "colony collapse disorder" - that phenomenon was a very specific suite of responses that results in colony depopulation and is not synonymous with honey bee losses. In fact, it has been >5 years since there have been notable reports of that phenotype.

Please consider all of the requests of the reviewers - I think these will improve the  manuscript. If you disagree with any, please justify in your response.

We would appreciate receiving your revised manuscript by Mar 20 2020 11:59PM. To enhance the reproducibility of your results, we recommend that if applicable you deposit your laboratory protocols in protocols.io, where a protocol can be assigned its own identifier (DOI) such that it can be cited independently in the future. For instructions see: http://journals.plos.org/plosone/s/submission-guidelines#loc-laboratory-protocols

We look forward to receiving your revised manuscript.

Kind regards,

Adam G Dolezal

Academic Editor

PLOS ONE

Journal Requirements:

2. Your ethics statement must appear in the Methods section of your manuscript. If your ethics statement is written in any section besides the Methods, please move it to the Methods section and delete it from any other section. Please also ensure that your ethics statement is included in your manuscript, as the ethics section of your online submission will not be published alongside your manuscript.

Reviewers' comments:

Reviewer's Responses to Questions

**Comments to the Author**

1. Is the manuscript technically sound, and do the data support the conclusions?

Reviewer #1: Yes

Reviewer #2: Yes

2. Has the statistical analysis been performed appropriately and rigorously? 

Reviewer #1: Yes

Reviewer #2: Yes

3. Have the authors made all data underlying the findings in their manuscript fully available?

Reviewer #1: Yes

Reviewer #2: No

4. Is the manuscript presented in an intelligible fashion and written in standard English?

Reviewer #1: Yes

Reviewer #2: Yes

5. Review Comments to the Author

Reviewer #1: Title: Citizen science improves our understanding of native pollinator management

Authors: Logan Appenfeller, Zsofia Szendrei, Sarah Lloyd

The authors of this paper have conducted a well replicated study examining the effects of farm management, especially tillage and practices related to soil, and how these practices effect the abundance of squash bees and other bees visiting cucurbits in Michigan by using citizen science data reported over three summers. Overall, they found a greater total number of squash bees reported than honey bees, bumble bees, and other bees. No till and reduced tillage practices supported greater numbers of squash bees than full till practices and the use of mulch had no effect on squash bee abundance. Tillage and tillage depth did not have an effect on honey bee and bumble bee abundance, however, other bees were significantly affected by these features with greater numbers of other bees in no till and reduced tillage farms compared to full till. The results of this study are pertinent as wild bees are in decline globally. It is essential to understand how farm management affects bees in order to create management practices to support greater bee communities.

General Comments: I commend the writer for a well written and easy to read paper. The topic was very interesting and pertinent. The graphs and figures all look very nice. See below for some suggestions to edit.

There was poor organization of the supplemental table making it a bit hard to digest

Inconsistent use of the oxford comma.

The use of a period after figure in the text and captions is inconsistent (e.g., fig 1, fig.1, fig.1.). Please check these throughout.

Line 39. The sentence that starts with “Management practices that increase…” you state that practices which increase bee abundance and richness can ameliorate pollinator deficiencies. However, I think I understand that you mean to say that increased pollinator abundance and richness will ameliorate crop pollination deficiencies, especially for crops that are more effectively pollinated by native bees.

Line 45. You explain that because native bees are hard to study many researchers are relying on citizen science data to map abundance and distributions of native bees, however, the reference [15] you provide is not for wild bee data. It may be better to use this reference more generally, stating that citizen science data allow researchers to obtain a greater quantity of samples from a larger geographic region, and then focus on more recent and bee related studies to support that this is being done with native bees. See some examples references:

Mason, L., & Arathi, H. (2019). Assessing the efficacy of citizen scientists monitoring native bees in urban areas. Global Ecology and Conservation, 17, E00561.

Lander, T. (2019). Network modelling, citizen science and targeted interventions to predict, monitor and reverse bee decline. Plants People Planet https://doi.org/10.1002/ppp3.10068

Graham, J., Tan, Q., Jones, L., & Ellis, J. (2014). Native Buzz: Citizen scientists creating nesting habitat for solitary bees and wasps. Florida Scientist, 77(4), 204-218. Retrieved January 7, 2020, from www.jstor.org/stable/24321925

Line 56. Italicize the genus Cucurbita

Lines 57-71. This paragraph focuses on explaining the potential for management practices to disturb squash bees and then outlines how previous research has found mixed results for various types of disturbances. The next paragraph on lines 72-78 discuss that you use citizen science to assess how land management in Michigan impacts squash bees. The link between previous work and why you are doing this study is missing. How will your study bridge the gap in knowledge and or clarify the already mixed results that exists on this topic?

Based on what I am reading I would guess that your reasoning is as follows: Multiple studies have assessed how various farm management practices impact the squash bee. However, many of these studies are finding mixed results, do not investigate more than one management practice in the same study, and often suffer from reduced sample sizes. In this study you attempt to clarify how farm management impacts squash bees by assessing multiple management practices such as soil tillage, depth, and mulch. You use citizen science to conduct this study because it will allow you to increase your sample size, therefore, strengthening the magnitude of your result, as well as educate the public on the importance of squash bees.

Line 82-83. In some cases you capitalize Master Gardeners and in others you spell it as Master gardeners. Please pick one and check it throughout.

Line 161. Hyphenate “Kruskal Wallis” as it is done in line 152

Line 217. For the sake of transparency, can you go ahead and just add the coordinating stat for each of the features listed (e.g., insecticides, irrigation, and vine crop)?

Paragraph beginning on Line 225. In your statistical analysis section you stated that you performed Dunn post hoc analysis when the overall Kruskal-Wallis test was significant to determine differences in honey bee, bumble bee, and other bees response to farm management, however, it does not appear that you report post hoc stats for any of these differences. Maybe I am misunderstanding this type of test, but it appears to me that you report the main test result and then you are simply stating differences based on the mean. I also do not see the post hoc differences listed in SI Table 1. Can you add the appropriate stats?

For the Discussion.

Seeing that you have taken the time to describe how mixed the results are in literature about how farm management affects squash bees, you may consider comparing the results of your study to these previous claims in addition to explaining the reasoning behind your results. I will leave this up to the authors.

If you are not over the word limit you may want to discuss the findings of the honey bee, bumble bee, and other bee data a bit more. As it stands now you have only 1 sentence (line 300) about this. It is not super clear to me what the benefit of measuring these farm practices was for other bees in reference to your study and how it relates to the squash bees. I think I know what you were going for but it is not clearly described. As it is now, it just seems like extra data that got tossed in and were never discussed. Given the differences in nesting and dietary habits, I think you may have more to elaborate on.

Line 299. The jump from geographic range to flower visitation in the next sentence confuses me. These two things are not the same. Could you discuss the geographic expansion compared to historical records? Are there any ideas as to why these expansions happened? Is it likely a product of the lack of people reporting or is it because of an increase in farming for Cucurbits by small farmers? My guess is that it is related to the lack of studies and reports by farmers, again highlighting the usefulness of this type of citizen science.

Reviewer #2: This study uses a citizen science approach to investigate how soil management in pumpkin and squash farms impact squash bee abundance across the state of Michigan (USA). I really enjoyed reading this manuscript and I think it provides an important contribution to the question of the impact of tillage on the abundance of squash bees in these cropping systems. I have several comments that I hope will help emphasize the important outcomes of the project.

(1) The title does not accurately describe the goal of the study. I suggest you replace “native bee pollinator management” for “the impact of soil management on wild pollinator abundance in agroecosystems”. This suggestion should be incorporated in other parts of the manuscript (e.g., L24, L52). This study really focuses on soil management, not general farm management.

(2) L36 - Colony collapse disorder is not the main cause of annual honey bee losses in North America (Kulhanek et al 2017). Losses are due to the interaction of multiple factors including poor nutrition, pest and pathogens, and pesticides.

(3) L42 - Why is it challenging to study native bees? Please explain.

(4) L45 - I suggest you cite Ryan et al. Proc Royal Soc B (2018). This opinion paper provides a nice review of the role of citizen science in answering questions in the context of agriculture.

(5) L56 - Cucurbits should be italicized

(6) L72 - Why is it important to spread awareness about squash bees? Please explain.

(8) The use of a citizen science (CS) approach for this project is one of the most interesting aspects of the study. However, after reading the introduction, I think it misses the opportunity to explain the challenges of using CS to collect data. L42-L54 explain the advantages of CS but not the limitations. I think explaining the limitations in the introduction will frame the results of this study into a better context (you received back 276 surveys!).

(9) L192 - Please provide a brief explanation of the difference between tillage and reduced tillage.

(10) L234 - Please provide the scientific name of “winter squash”

(11) A general comment about the interpretation of these results. In this study, the authors find that tillage has a negative effect on squash bee visitation (presumably abundance) in Cucurbits farms. However, when the impact of soil tillage on squash bee populations has been experimentally tested, the results are mixed usually indicating a lack of effect. Is it possible that tillage is correlated with other types of management practices that are directly driving the changes in squash bee populations? For example, is tillage and crop rotation correlated? I don’t know if the authors collected that type of data, but I think it would be worth mentioning this (or other confounding factors) in the discussion to try to reconcile the conflicting results of multiple studies.

6. PLOS authors have the option to publish the peer review history of their article (what does this mean?). If published, this will include your full peer review and any attached files.

Reviewer #1: No

Reviewer #2: No

---

## [Author Response · Author response to Decision Letter 0]

17 Feb 2020

Dear Editor,

Please find our responses to the reviewers’ comments below. We addressed each of the comments line by line as requested and made several changes as a result throughout the text. Most of these were relatively minor; the most notable changes were in the discussion where we added text in several places to address the comment on the lack of discussion regarding our findings related to the geographic range and other bee groups. The suggestions were helpful in improving the manuscript overall and we are looking forward to hearing from you.

Best,

The authors

Academic Editor: 

Journal Requirements:

Response: We now include a superscript to indicate university affiliations in the author byline, and files for re-submission have been named according to instructions provided by the Academic Editor in the decision email. 

2. Your ethics statement must appear in the Methods section of your manuscript. If your ethics statement is written in any section besides the Methods, please move it to the Methods section and delete it from any other section. Please also ensure that your ethics statement is included in your manuscript, as the ethics section of your online submission will not be published alongside your manuscript.

Response: We now provide our ethics statement in the Methods section. L-153-155 now reads: 

“IRB Number is x17-688e; i054192; this survey was deemed exempt and was not subjected to review by an institutional review board or ethics committee.”

Reviewer 1:

Comment: There was poor organization of the supplemental table making it a bit hard to digest.

Inconsistent use of the oxford comma. The use of a period after figure in the text and captions is inconsistent (e.g., fig 1, fig.1, fig.1.). Please check these throughout.

 Response: The supplemental table has been simplified and re-arranged to reduce its size and make it easier to interpret. 

Commas have been added to sentences starting at lines 32, 93, 284, 301, and 328 to make use of the oxford comma consistent. These sentences now read, respectively:

“Pollinators are important in both natural and managed ecosystems to maintain plant genetic diversity, contribute to ecosystem stability, and sustain crop production [1].”

“To train Master Gardeners we invited them to webinar presentations held in June 2017 and 2018 where the methods, project goals, and preliminary results were discussed.”

“Some of the successes of our project are due to identifying an appropriate target audience for involvement with the project, a thorough volunteer education process, simplicity of the survey protocol, and ease of data submission.”

“To keep the survey relatively short, we omitted some questions that would have provided us with valuable data, for example, recording soil-type, time of day, weather conditions, flower sex, and flower abundance would have allowed us to answer additional questions.”

“Strip-tillage is often accompanied by the presence of cover crop residues (mulch) between strips of tilled soil which can help maintain soil moisture, reduce soil erosion, and inhibit weed seed germination [56].”

Figure references in Lines 123 (“S1 Fig.”), 140 (“Fig. 2.”), 167 (“S1 Fig.”), and 355 (“Fig. 4”) have been corrected to “S1 Fig”, “Fig 2.”, “S1 Fig”, and “Fig 4” respectively. “Fig 2.” in line 140 is followed by a period because it starts a figure legend. 

Comment: Line 39. The sentence that starts with “Management practices that increase…” you state that practices which increase bee abundance and richness can ameliorate pollinator deficiencies. However, I think I understand that you mean to say that increased pollinator abundance and richness will ameliorate crop pollination deficiencies, especially for crops that are more effectively pollinated by native bees.

 Response: We agree with the reviewer’s suggestion to increase the clarity of this sentence. The sentence (L41) has been modified to: 

“Management practices that increase abundance and species richness of native bees can ameliorate crop pollination deficiencies [9], especially for crops that are more effectively pollinated by native bees [10–15].”

Comment: Line 45. You explain that because native bees are hard to study many researchers are relying on citizen science data to map abundance and distributions of native bees, however, the reference [15] you provide is not for wild bee data. It may be better to use this reference more generally, stating that citizen science data allow researchers to obtain a greater quantity of samples from a larger geographic region, and then focus on more recent and bee related studies to support that this is being done with native bees. See some examples references: 

Mason, L., & Arathi, H. (2019). Assessing the efficacy of citizen scientists monitoring native bees in urban areas. Global Ecology and Conservation, 17, E00561.

Lander, T. (2019). Network modelling, citizen science and targeted interventions to predict, monitor and reverse bee decline. Plants People Planet https://doi.org/10.1002/ppp3.10068

Graham, J., Tan, Q., Jones, L., & Ellis, J. (2014). Native Buzz: Citizen scientists creating nesting habitat for solitary bees and wasps. Florida Scientist, 77(4), 204-218. Retrieved January 7, 2020, from www.jstor.org/stable/24321925

 Response: The reviewer makes a good point here and we have modified the text to incorporate two new references [16,17] which mention challenges of studying changes to insect populations/distributions. We also now refer to the Gardiner et al. [18] study to provide a more general example of how citizen science provides a cost effective means of collecting large amounts of insect data across broad geographic areas. We then provide more specific examples of how citizen science has been demonstrated to be an effective tool for native bee monitoring, incorporating the 3 references suggested by the reviewer. The text (L44-59) now reads:

“Studying changes in insect populations is often challenging [16,17], and in order to collect baseline abundance and distribution data, insect monitoring has in some cases turned to citizen science as an effective method for gathering large datasets across broad geographic areas with low costs compared to traditional methods [18,19]. Although citizen science can suffer from limitations such as data accuracy and participant retention, these issues can be negated with proper planning and participant training as demonstrated by many successful citizen science projects. For example, citizen science can be an effective method for monitoring native bees [20–22]. Citizen scientist observations can describe bee species dynamics as well as specimens collected by professional researchers [23], provide data on specific aspects of bee biology, including the nesting habits of solitary bee species [24], and the impacts of flowers and surrounding natural land cover on plant-bee interactions [25]. Participants in pollinator citizen science projects often volunteer because of a desire to learn about bees and to contribute to science [26]. This provides opportunities for large-scale, cost-effective studies that simultaneously allow scientists to educate the public about ecological issues such as the loss of biodiversity. Actively engaging with the public through hands-on experiences provides more impactful education that can enhance learning and inspire continued action [27].”

Comment: Line 56. Italicize the genus Cucurbita. 

 Response: Cucurbita is now italicized (L61). 

Comment: Lines 57-71. This paragraph focuses on explaining the potential for management practices to disturb squash bees and then outlines how previous research has found mixed results for various types of disturbances. The next paragraph on lines 72-78 discuss that you use citizen science to assess how land management in Michigan impacts squash bees. The link between previous work and why you are doing this study is missing. How will your study bridge the gap in knowledge and or clarify the already mixed results that exists on this topic? Based on what I am reading I would guess that your reasoning is as follows: Multiple studies have assessed how various farm management practices impact the squash bee. However, many of these studies are finding mixed results, do not investigate more than one management practice in the same study, and often suffer from reduced sample sizes. In this study you attempt to clarify how farm management impacts squash bees by assessing multiple management practices such as soil tillage, depth, and mulch. You use citizen science to conduct this study because it will allow you to increase your sample size, therefore, strengthening the magnitude of your result, as well as educate the public on the importance of squash bees.

 Response: The reviewer provides an excellent point here and we have added to the text to clarify the relevance of our study. L77-88 in the revised manuscript now read: 

“Although multiple studies have examined the impacts of farm management on squash bees, the scope of investigation has often been limited to one management practice at a time, sample sizes have been relatively low, and results have often been mixed [31–34]. Here, we used a citizen science survey to determine how squash bee abundance varies according to multiple farm management practices including tillage type, depth, and mulch, and ascertain the distribution of squash bees in Michigan. Citizen science allowed us to increase sampling while providing opportunities to spread awareness among the public about the importance of squash bees which may pollinate about two-thirds of squash grown commercially in the United States [36]. Previous research indicates that citizen science projects are more successful if the participants have prior interest in the subject matter [26,37,38] thus we recruited Michigan State University (MSU) Extension Master Gardeners because of their interest in agriculture, their level of scientific knowledge, and their commitment to educate others in their communities.”

Comment: Line 82-83. In some cases you capitalize Master Gardeners and in others you spell it as Master gardeners. Please pick one and check it throughout. 

 Response: “Master Gardener” is now used throughout the text. The title is capitalized because we are referring specifically to Michigan State University Extension Master Gardeners. 

Comment: Line 161. Hyphenate “Kruskal Wallis” as it is done in line 152.

 Response: The test name has been corrected to Kruskal-Wallis (L182). 

Comment: Line 217. For the sake of transparency, can you go ahead and just add the coordinating stat for each of the features listed (e.g., insecticides, irrigation, and vine crop)?

 Response: We now include each of the coordinating statistical results for insecticides, irrigation, vine crop observed, and area devoted to cucurbits in the body of the text (L237-241) which now reads:

 “Squash bee visitation was not affected by insecticides (�2 = 2.53, df = 2, p = 0.28), irrigation (�2 = 3.97, df = 2, p = 0.14), the type of vine crop observed (�2 = 5.68, df = 3, p = 0.13), or amount of area devoted to cucurbit growth (�2 = 2.10, df = 1, p = 0.15).”

Comment: Paragraph beginning on Line 225. In your statistical analysis section you stated that you performed Dunn post hoc analysis when the overall Kruskal-Wallis test was significant to determine differences in honey bee, bumble bee, and other bees response to farm management, however, it does not appear that you report post hoc stats for any of these differences. Maybe I am misunderstanding this type of test, but it appears to me that you report the main test result and then you are simply stating differences based on the mean. I also do not see the post hoc differences listed in SI Table 1. Can you add the appropriate stats?

 Response: The reviewer is correct in that Dunn’s test results were not originally reported in the text but rather the Kruskal-Wallis results alone. Significant pairwise differences between factor levels were calculated using Dunn’s test and are indicated by different letters following mean ± SEM values in S1 Table. Nonetheless, we have added both the non-significant Kruskal-Wallis test results and the Dunn’s test statistics to this relevant section throughout to add transparency and clarity. L245-279 now read:

“No relationship was observed between honey bee visitation and tillage type (�2 = 1.40, df = 2, p = 0.50), tillage depth (�2 = 1.05, df = 2, p = 0.59), insecticides (�2 = 1.13, df = 2, p = 0.57), irrigation (�2 = 0.30, df = 2, p = 0. 86), or the type of vine crop observed (�2 = 6.50, df = 3, p = 0.09; S1 Table). However, mulch affected honey bee visitation (�2 = 12.02, df = 3, p = 0.01). The mean number of honey bees reported in farms using plant material mulch (mean = 0.44 � 0.06 (SE)) was more than 3 times greater than when mulch (mean = 0.13 � 0.04 (SE)) was not present (p < 0.01). No other significant differences were observed among mulch types, however, the mean number of honey bees observed when plant material and plastic (mean = 0.33 � 0.14 (SE)) or plastic alone (mean = 0.40 � 0.40 (SE)) were present was approximately 2.5-3 times greater than when mulch was not present (S1 Table). Average honey bee visitation was also 2 times greater when cucurbit area was greater than 0.4 hectares (mean = 0.64 � 0.24 (SE)) compared to less than 0.4 hectares (mean = 0.31 � 0.04 (SE); �2 = 2.92, df = 1, p = 0.09; S1 Table). No relationship was observed between bumble bee visitation and tillage type (�2 = 2.96, df = 2, p = 0.23), tillage depth (�2 = 5.05, df = 2, p = 0.08), mulch (�2 = 1.56, df = 3, p = 0.67), insecticides (�2 = 3.62, df = 2, p = 0.16), or irrigation (�2 = 4.13, df = 2, p = 0.13). Interestingly, bumble bee visitation was 3.5 times greater when cucurbit area was less than 0.4 hectares (mean = 0.54 � 0.05 (SE)) compared to greater than 0.4 hectares (mean = 0.15 � 0.10 (SE); �2 = 3.60, df = 1, p = 0.06). Bumble bee visitation also varied by the type of vine crop observed (�2 = 20.56, df = 3, p < 0.01). The mean number of bumble bees observed visiting winter squash varieties (C. pepo, mean = 0.80 � 0.09 (SE)) was more than two times greater than when mixed varieties (combination of C. pepo and Cucumis; mean = 0.35 � 0.12 (SE); p < 0.01) or summer squash varieties alone (C. pepo, mean = 0.36 � 0.06 (SE); p < 0.01) were observed. No other significant differences were found among vine crop types; however, the mean number of bumble bee visits in surveys where summer and winter squash flowers were observed together (mean = 0.70 � 0.18 (SE)) was approximately 2 times greater than when mixed or summer squash flowers alone were observed. Other bee visitation did not vary by mulch (�2 = 4.26, df = 3, p = 0.23), insecticides (�2 = 4.77, df = 2, p = 0.09), irrigation (�2 = 0.58, df = 2, p = 0.75), or the type of vine crop observed (�2 = 6.07, df = 3, p = 0.11; S1 Table). Although, visitation by other bees varied according to tillage type (�2 = 11.04, df = 2, p < 0.01) with the mean number of other bees observed in farms using no tillage (mean = 1.33 � 0.14 (SE)) and full tillage (mean = 1.36 � 0.32 (SE)) being approximately 2 times greater than in farms using reduced tillage (mean = 0.68 � 0.09 (SE); p < 0.01; p = 0.16). Tillage depth also affected other bee visitation (�2 = 8.66, df = 2, p = 0.01) with the mean number of other bee visits in farms using no tillage (mean = 1.33 � 0.14 (SE)) being around 1.5 times greater than in farms with 3-15 cm tillage depth (mean = 0.78 � 0.10 (SE); p = 0.01) or 15-25 cm (mean = 0.96 � 0.24 (SE); p = 0.71; S1 Table).”

We also modified a sentence in the methods section specifying that Dunn’s test was used to determine pairwise differences among factor levels only for factors with more than 2 levels. When there are only 2 levels within a factor (e.g. Cucurbit Area), the Kruskal-Wallis equates to a pairwise comparison between the 2 levels. This section (L181-185) now reads:

“When Kruskal-Wallis tests were significant, post hoc analyses were conducted using the ‘dunn.test’ function in the ‘dunn.test’ [48] package with the ‘holm’ p-adjustment method to control family-wise error rates, to determine differences in bee abundances among groups for factors with more than 2 groups (� < 0.05).” 

Furthermore, a sentence has been added to the legend for S1 Table which reads: “Significant pairwise differences among factor levels were determined using Dunn’s test and are indicated by different letters following mean � SEM values (� < 0.05).” The first place that these letters appear is for Other Bees in the Tillage Type section of S1 Table. 

Comment: For the Discussion. Seeing that you have taken the time to describe how mixed the results are in literature about how farm management affects squash bees, you may consider comparing the results of your study to these previous claims in addition to explaining the reasoning behind your results. I will leave this up to the authors. If you are not over the word limit you may want to discuss the findings of the honey bee, bumble bee, and other bee data a bit more. As it stands now you have only 1 sentence (line 300) about this. It is not super clear to me what the benefit of measuring these farm practices was for other bees in reference to your study and how it relates to the squash bees. I think I know what you were going for but it is not clearly described. As it is now, it just seems like extra data that got tossed in and were never discussed. Given the differences in nesting and dietary habits, I think you may have more to elaborate on.

 Response: We now provide new discussion pertaining to observations for the other categories of bees and speculate about observed patterns and/or lack thereof. L353-366 now reads: 

“In our survey, squash bees were observed visiting flowers about 3-4 times more often than honey bees, bumble bees, or other bees, comprising more than 50% of the total number of bees reported (Fig 4). We did not expect that honey bees would respond to mulching or tillage because they are not ground nesting bees and therefore do not directly interact with the soil. We observed a positive effect of plant mulches on honey bee abundance which may be due to an indirect impact of soil management practices on these bees through affecting plant health or flower abundance. Bumble bees did not respond to ground management practices, and although they are ground nesting bees, they can cover long distances during foraging and are likely nesting outside of squash fields [32,61–63]. The overall lack of response by honey and bumble bees to most soil management practices could also be because they are dietary generalists feeding on other available sources of pollen and nectar [64]. Additionally, honey bees in particular visit squash flowers primarily for nectar as indicated by their preference for pistillate squash flowers [65]. Therefore, squash flower abundance, quality, and/or field attributes dictated by soil management may be less consequential for these bees.”

Comment: Line 299. The jump from geographic range to flower visitation in the next sentence confuses me. These two things are not the same. Could you discuss the geographic expansion compared to historical records? Are there any ideas as to why these expansions happened? Is it likely a product of the lack of people reporting or is it because of an increase in farming for Cucurbits by small farmers? My guess is that it is related to the lack of studies and reports by farmers, again highlighting the usefulness of this type of citizen science. 

 Response: We have taken the reviewer’s suggestions into account and now elaborate more over this feature. L347-352 now reads: 

“Citizen science was also an effective means of examining the current geographic range of the squash bee because of the relatively broad participation in our study. Citizen scientists reported 7 new county records for this species, and while geographic range expansion may be responsible for such patterns, we hypothesize that a more likely explanation is a lack of historical reports and/or an increase in the number of small organic farms in Michigan which more often practice farm management methods that can promote native bees [58–60].”

 

Reviewer 2: 

Comment: (1) The title does not accurately describe the goal of the study. I suggest you replace “native bee pollinator management” for “the impact of soil management on wild pollinator abundance in agroecosystems”. This suggestion should be incorporated in other parts of the manuscript (e.g., L24, L52). This study really focuses on soil management, not general farm management.

 Response: The title has been changed to: 

 “Citizen science improves our understanding of the impact of soil management on wild pollinator abundance in agroecosystems”

Comment: (2) L36 - Colony collapse disorder is not the main cause of annual honey bee losses in North America (Kulhanek et al 2017). Losses are due to the interaction of multiple factors including poor nutrition, pest and pathogens, and pesticides.

 Response: We thank the reviewer for pointing this out. To improve accuracy and clarity the sentence (L36-39) has been amended to:

“Honey bees are the most prolific pollinators of pollinator dependent crops [3], however annual losses of managed honey bees can currently reach as high as 50% due to a suite of factors such as exposure to pesticides, reduced forage availability, parasites, and diseases [4,5].”

Comment: (3) L42 - Why is it challenging to study native bees? Please explain.

 Response: This section has been changed to first assert that studying changes in insect populations more broadly speaking presents challenges (references provided) and how citizen science has emerged as an efficient method in this context. We then transition into examples of citizen science projects that specifically focused on native bees. L44-54 now reads:

 “Studying changes in insect populations is often challenging [16,17], and in order to collect baseline abundance and distribution data, insect monitoring has in some cases turned to citizen science as an effective method for gathering large datasets across broad geographic areas with low costs compared to traditional methods [18,19]. Although citizen science can suffer from limitations such as data accuracy and participant retention, these issues can be negated with proper planning and participant training as demonstrated by many successful citizen science projects. For example, citizen science can be an effective method for monitoring native bees [20–22]. Citizen scientist observations can describe bee species dynamics as well as specimens collected by professional researchers [23], provide data on specific aspects of bee biology, including the nesting habits of solitary bee species [24], and the impacts of flowers and surrounding natural land cover on plant-bee interactions [25].” 

Comment: (4) L45 - I suggest you cite Ryan et al. Proc Royal Soc B (2018). This opinion paper provides a nice review of the role of citizen science in answering questions in the context of agriculture.

Response: We thank the reviewer for bringing this article to our attention. It is now referenced where suggested (L47). 

Comment: (5) L56 - Cucurbits should be italicized. 

 Response: Cucurbita is now italicized (L61).

Comment: (6) L72 - Why is it important to spread awareness about squash bees? Please explain.

 Response: We have added text and a reference to add to the our argument for the agricultural importance of squash bees. The text (L82-84) now reads: 

 “Citizen science allowed us to increase sampling while providing opportunities to spread awareness among the public about the importance of squash bees which may pollinate about two-thirds of squash grown commercially in the United States [36].”

Comment: (8) The use of a citizen science (CS) approach for this project is one of the most interesting aspects of the study. However, after reading the introduction, I think it misses the opportunity to explain the challenges of using CS to collect data. L42-L54 explain the advantages of CS but not the limitations. I think explaining the limitations in the introduction will frame the results of this study into a better context (you received back 276 surveys!).

Response: We now briefly address potential limitations of citizen science in the introduction before giving examples of successful projects. We further elaborate on in the discussion (limitations were already mentioned in the discussion of the previously submitted manuscript). L47-49 now reads: 

 “Although citizen science can suffer from limitations such as data accuracy and participant retention, these issues can be negated with proper planning and participant training as demonstrated by many successful citizen science projects.”

Comment: (9) L192 - Please provide a brief explanation of the difference between tillage and reduced tillage.

Response: A brief description is now provided in the methods to clarify the difference between tillage types. L124-132 now read: 

 “Tillage type was one of the primary factors of interest in our study and participants could select no tillage, reduced tillage, or full tillage. No tillage is characterized by a lack of soil disturbance between harvesting and planting crops resulting in the presence of crop stubble or residues. Reduced tillage (a.k.a. conservation tillage) is defined by lower tillage intensity resulting in the retention of some crop residues on the soil surface. Both of these methods help to prevent soil erosion, increase water retention, and conserve energy resources. Full tillage (a.k.a. conventional tillage) uses cultivation (e.g. ploughing, harrowing) as the primary means of weed control and seedbed preparation resulting in a loose soil surface and lack of plant residues on the soil surface [41].”

Comment: (10) L234 - Please provide the scientific name of “winter squash”

 Response: We provide the species name for the winter squash varieties we refer to. L262-266 now reads:

 “The mean number of bumble bees observed visiting winter squash varieties (C. pepo, mean = 0.80 � 0.09 (SE)) was more than two times greater than when mixed varieties (combination of C. pepo and Cucumis; mean = 0.35 � 0.12 (SE); p < 0.01) or summer squash varieties alone (C. pepo, mean = 0.36 � 0.06 (SE); p < 0.01) were observed.”

Comment: (11) A general comment about the interpretation of these results. In this study, the authors find that tillage has a negative effect on squash bee visitation (presumably abundance) in Cucurbits farms. However, when the impact of soil tillage on squash bee populations has been experimentally tested, the results are mixed usually indicating a lack of effect. Is it possible that tillage is correlated with other types of management practices that are directly driving the changes in squash bee populations? For example, is tillage and crop rotation correlated? I don’t know if the authors collected that type of data, but I think it would be worth mentioning this (or other confounding factors) in the discussion to try to reconcile the conflicting results of multiple studies.

Response: We now more fully address this concern noting that although it is possible that other factors may have effects on squash bee abundance, soil management likely has the strongest impacts. L318-327 now reads: 

“It is possible that tillage is correlated with other types of management practices that are responsible for changes in squash bee abundance, such as crop rotation or insecticide use. Considering that squash bees nest close to their host plants [33] and that Cucurbita crops are typically rotated, the number of squash bees visiting flowers is likely influenced by the management of previous year’s fields, and the distance between these and current plantings. This highlights the relevance of ground management not only within individual fields but at the farm level. However, previous research demonstrating significant impacts on squash bee abundance due to soil management combined with a lack of observed impacts on generalist pollinators like honey bees and bumble bees suggests that although other forms of farm management may have some impact on squash bees, soil tillage is likely to impose strong effects [32].”

---

## [Editor Report · Decision Letter 1]

20 Feb 2020

Citizen science improves our understanding of the impact of soil management on wild pollinator abundance in agroecosystems

PONE-D-19-34559R1

Dear Dr. Appenfeller,

We are pleased to inform you that your manuscript has been judged scientifically suitable for publication and will be formally accepted for publication once it complies with all outstanding technical requirements.

With kind regards,

Adam G Dolezal

Academic Editor

PLOS ONE

Additional Editor Comments (optional):

Thank you for your thorough response to the reviewers; I think the changes have improved the preset nation of your study.

One thing I do want to note is I see a change in author order. I flag this just to make sure it was intentional. If it was not, please let me/the journal know ASAP. If it is now as you intend it, it should be fine.
---

## [Editor Report · Acceptance letter]

26 Feb 2020

PONE-D-19-34559R1 

Citizen science improves our understanding of the impact of soil management on wild pollinator abundance in agroecosystems 

Dear Dr. Appenfeller:

I am pleased to inform you that your manuscript has been deemed suitable for publication in PLOS ONE. Congratulations! Your manuscript is now with our production department. 

With kind regards,

on behalf of

Dr. Adam G Dolezal 

Academic Editor

PLOS ONE